



# Arctic black carbon during PAMARCMiP 2018 and previous aircraft experiments in spring

Sho Ohata[1,2,*], Makoto Koike[3,*], Atsushi Yoshida[3,4], Nobuhiro Moteki[3], Kouji Adachi[5], Naga Oshima[5],
Hitoshi Matsui[6], Oliver Eppers[7,8], Heiko Bozem[7], Marco Zanatta[9,10], and Andreas B. Herber[9]

[1]Institute for Space–Earth Environmental Research, Nagoya University, Nagoya, Aichi, Japan
[2]Institute for Advanced Research, Nagoya University, Nagoya, Aichi, Japan
[3]Department of Earth and Planetary Science, Graduate School of Science, University of Tokyo, Tokyo, Japan.
[4]National Institute of Polar Research, Tokyo, Japan
[5]Meteorological Research Institute, Tsukuba, Japan
[6]Graduate School of Environmental Studies, Nagoya University, Nagoya, Japan.
[7]Johannes Gutenberg University of Mainz, Institute for Atmospheric Physics, Mainz, Germany
[8]Max Planck Institute for Chemistry, Particle Chemistry Department, Mainz, Germany
[9]Alfred Wegener Institute Helmholtz Centre for Polar and Marine Research (AWI), Bremerhaven, Germany
[10]LISA, UMR CNRS 7583, Université Paris-Est-Créteil, IPSL, Créteil, France

[*]These authors contributed equally to this work.

*Correspondence to*: Sho Ohata (sho.ohata@isee.nagoya-u.ac.jp)

Running title: Arctic BC during PAMARCMiP 2018



**Abstract.**

Vertical profiles of the mass concentration of black carbon (BC) were measured at altitudes up to 5 km during the PAMARCMiP aircraft-based field experiment conducted around the Northern Greenland Sea (Fram Strait) during March and April 2018, with operation base Station Nord (81.6°N, 16.7°W). Median BC mass concentrations in individual altitude ranges were 7–18 ng m$^{-3}$ at standard temperature and pressure at altitudes below 4.5 km. These concentrations were systematically lower than previous

observations in the Arctic in spring conducted by ARCTAS-A in 2008 and NETCARE in 2015 and similar to those observed during HIPPO3 in 2010. Column amounts of BC for altitudes below 5 km in the Arctic (>66.5°N, $COL_{BC}$), observed during the ARCTAS-A and NETCARE experiments were higher by factors of 4.2 and 2.7, respectively, than those of the PAMARCMiP experiment. These differences could not be explained solely by the different locations of the experiments. The year-to-year

variation of $COL_{BC}$ values generally corresponded to that of biomass burning activities in northern high latitudes over western and eastern Eurasia. Furthermore, numerical model simulations estimated the year-to-year variation of contributions from anthropogenic sources to be smaller than 30–40%. These results suggest that the year-to-year variation of biomass burning activities likely affected BC amounts in the Arctic troposphere in spring, at least in the years examined in this study. The year-to-year

variations in BC mass concentrations were also observed at the surface at high Arctic sites Ny-Ålesund and Barrow, although their magnitudes were slightly lower than those in $COL_{BC}$.

Numerical model simulations in general successfully reproduced the observed $COL_{BC}$ values for PAMARCMiP and HIPPO3 (within a factor of 2), whereas they markedly underestimated the values for ARCTAS-A and NETCARE by factors of 3.7–5.8 and 3.3–5.0, respectively. Because anthropogenic

contributions account for nearly all of the $COL_{BC}$ (82 – 98%) in PAMARCMiP and HIPPO3, the good agreements between the observations and calculations for these two experiments suggest that anthropogenic contributions were generally well reproduced. However, the significant underestimations of $COL_{BC}$ for ARCTAS-A and NETCARE suggest that biomass burning contributions were underestimated.

In this study, we also investigated plumes with enhanced BC mass concentrations, which were affected by biomass burning emissions, observed at 5 km altitude. Interestingly, the mass-averaged diameter of BC (core) and the shell-to-core diameter ratio of BC-containing particles in the plumes were generally not very different from those in other air sampled, which were considered to be mostly aged anthropogenic BC. These observations provide useful bases to evaluate numerical model

simulations of the BC radiative effect in the Arctic region in spring.



## 1 Introduction

Over the past few decades, the annual average temperature in the Arctic has increased almost twice as fast as it has elsewhere in the world (IPCC, 2013). The main driver of this warming is the global increase in the concentration of carbon dioxide ($CO_2$). However, various other climate forcers and feedback processes amplify warming in the Arctic (e.g., Serreze and Barry, 2011). Black carbon (BC) particles in the Arctic have been intensively studied as one of the Short-Lived Climate Forcers (SLCFs)

because reductions in their emissions could reduce positive radiative forcing with much shorter timescales as compared with reductions of $CO_2$ (Arctic Monitoring and Assessment Programme (AMAP), 2015).

Most of the Arctic BC is considered to be transported from regions outside the Arctic, although gas flaring and some other small emission sources within the Arctic also make some contributions (Stohl et

al., 2013). Previous studies using numerical model simulations indicate that contributions of BC emission sources change with season, year, and locations within the Arctic. Xu et al. (2017) reported that fossil fuel and biofuel anthropogenic (AN) emissions dominated at all levels (>90%) for the high Arctic region during spring in 2009, 2011, and 2015 when aircraft experiments were conducted. Wang et al. (2011) reported that open biomass burning (BB) accounted for 50% of the total BC in the Arctic

(>60°N) tropospheric column in April 2008, a year in which BB activity was quite intense (Warneke et al., 2009; Evangeliou et al., 2016). Considering the year-to-year variation of BB activities and uncertainties in AN and BB BC emissions, estimates of the relative contributions of various sources to Arctic BC still have considerable uncertainties. Furthermore, possible inter-annual increases in BB activities in the warming boreal forest in the near future may further increase the BB BC emissions.

Measurements of vertical profiles of Arctic BC are important for the following two reasons. First, vertical profiles of BC generally reflect BC sources and transport/wet removal processes, thus they provide useful information for investigating these sources and atmospheric processes. Air parcels originating at lower latitudes are uplifted along isentropic surfaces when they are transported to the Arctic, and some BC is removed by precipitation in association with uplifting (Shindell et al., 2008;

Matsui et al., 2011; Schulz et al., 2019).

Second and more importantly, the magnitude of the response to the surface temperature depends on the altitude range in which the light absorbing aerosols are located (Flanner 2013; Samset et al., 2013). Strong absorption of solar radiation by BC near the ground heats the air near the ground surface. This heating can reduce the cloud fraction, leading to an increase in the downward shortwave radiation flux

at the ground surface. BC at higher altitudes could cause weak surface warming or cooling through decreased surface insolation because the high stability of the Arctic atmosphere inhibits thermal mixing





with surface air. In addition, because BC deposition on snow and ice surfaces lowers their albedo and potentially accelerates ice–albedo feedback (Flanner et al., 2007; Tuzet et al., 2017), it is important to evaluate the altitude dependence of the wet removal processes of BC (Mori et al., 2020).

Aircraft measurements of Arctic BC were made by the following previous experiments in the spring season when direct radiative forcing of BC in the Arctic is considered to be largest: ARCPAC, ARCTAS-A, and POLARCAT in 2008 (Spackman et al., 2010; Matsui et al., 2011; Wang et al., 2011), PAMARCMiP in 2009 and in 2011 (Stone et al., 2010; Herber et al., 2012), HIPPO3 in 2010 (Schwarz et al., 2013), and NETCARE in 2015 (Schulz et al., 2019; Kodros et al., 2018). From these

measurements vertical profiles of BC mass concentrations, size distributions, and mixing states, as well as source contributions, transport pathways, and loss mechanisms were studied. However, measurements of BC vertical profiles are still limited in the Arctic and case studies were made for individual experiments.

The Polar Airborne Measurements and Arctic Regional Climate Model simulation Project

(PAMARCMiP) 2018 aircraft-based field experiment was carried out around the Northern Greenland Sea (Fram Strait) from 23 March to 4 April 2018 using Station Nord (81.6°N, 16.7°W) as the operation base (Herber et al., 2012, 2019; Donth et al., 2020; Koike et al., in press), as part of the ArctiC Amplification: Climate Relevant Atmospheric and SurfaCe Processes, and Feedback Mechanisms or (AC)[3] project (Wendisch et al., 2017). In this paper, we report vertical profiles of BC mass

concentrations, size distributions, and mixing states measured using a University of Tokyo single-particle soot photometer (Yoshida et al., 2020). The measurements (hereafter denoted as PAMARCMiP data) were compared with those made during these previous spring season experiments: ARCTAS-A in 2008, HIPPO3 in 2010, and NETCARE in 2015 (hereafter denoted as ARCTAS, HIPPO, and NETCARE). The year-to-year variation was examined from the viewpoint of BB activities.

## 115 2 Measurement

### 2.1 Instruments and aircraft experiment

In this study, BC was measured on board of a research aircraft by using a single-particle soot photometer (SP2; Droplet Measurement Technologies (DMT), Inc., Longmont, CO, USA) calibrated and operated by the University of Tokyo (Yoshida et al., 2020). Detailed descriptions of the SP2,

including calibration methods, are given elsewhere (e.g., Moteki and Kondo, 2010). Briefly, the SP2 uses the laser-induced incandescence technique and detects BC on a single-particle basis. It measured BC size distributions in the mass-equivalent diameter ($D_m$) range of approximately 70–850 nm. The void-free density of BC of 1.8 g cm$^{-3}$ was assumed to convert a particle mass to a mass equivalent diameter. The SP2 was calibrated using fullerene soot particles (Alfa Aeser, stock #40971, lot

#FS12S011, Moteki and Kondo, 2010; Kondo et al., 2011b). BC mass concentration ($M_{BC}$) values



reported in this paper were obtained by integrating the mass of BC particles within the detectable diameter range and are expressed in units of ng m$^{-3}$ under standard temperature and pressure (STP, 0ºC and 1013.25 hPa) conditions. The accuracy of the $M_{BC}$ measurements estimated from the uncertainty of the calibration and operational conditions of SP2 was about 10%.

As described in Appendix A, $M_{BC}$ measurements made with the University of Tokyo SP2 during ARCTAS (Sahu et al., 2012) were compared with NOAA SP2 measurements (NOAA-ARCPAC experiment, Spackman et al., 2010) and they agreed within 20%. The systematically higher U. Tokyo values might partly be attributed to a wider size range of U. Tokyo BC measurements and a difference in corrections for undetected BC between the measurements. Because $M_{BC}$ measurements were
consistently and traceably made by the individual groups, the systematic differences in absolute $M_{BC}$ values are expected to be smaller than 20% at least for PAMARCMiP and ARCTAS datasets (U. Tokyo) and HIPPO dataset (NOAA).

Aerosols were also sampled on filters onboard the aircraft during PAMARCMiP and analyzed using transmission electron microscopy (TEM, Adachi et al., 2021). In this study, we used potassium-
containing particles as an indicator of influences from biomass burning.

Gaseous carbon monoxide (CO) concentration was measured using an Aerolaser AL5002 CO monitor (Gerbig et al., 1999; Scharff et al., 2012), which detects fluorescence in vacuum ultraviolet radiation. $CO_2$ concentration was measured using an LI-7200 gas analyzer (LI-COR Ltd., USA; Burba et al., 2010; Lampert et al., 2018), which employs the non-dispersive infrared spectroscopy method.

For the aerosol measurements, ambient air was drawn into the aircraft cabin using a forward facing iso-kinetic inlet with a tip hole diameter of 4.6 mm designed and made by DMT, Inc. The sample air speed at the inlet tip and true air speed of the aircraft at various altitudes agreed within 10%, indicating that iso-kinetic sampling was achieved during the experiment. For the gas measurements, two backward facing sampling inlets (Ehrlich et al., 2019) were used.

PAMARCMiP aircraft experiment was carried out with the Alfred Wegener Institute (AWI) research aircraft Polar 5 (Wesche et al., 2016). The flight tracks are shown in Fig. 1 and the flights are listed in Table 1. The data obtained during 5 min after taking off and 5 min before landing were not used to avoid possible influences of local pollutants emitted from the base Station Nord. Vertical profiles of BC and other atmospheric species were measured up to 5 km during most of the research flights. Appendix
B presents 6-day kinematic back trajectories of air parcels measured onboard the aircraft.

## 2.2 Ground-based $M_{BC}$ measurements at Ny-Ålesund and Barrow

Along with year-to-year variations observed during aircraft experiments, those observed from ground-based $M_{BC}$ measurements at the high Arctic sites Ny-Ålesund in Svalbard (78.9°N, 11.9°E) and Barrow in Alaska (71.3°N, 156.6°W) are also presented for comparison in this paper. At both sites, $M_{BC}$ data



obtained by a continuous soot monitoring system called COSMOS (Kanomax, Osaka, Japan, Miyazaki
       et al., 2008; Kondo et al., 2009, 2011b) were used for the years 2015 and 2018. This filter-based
       absorption photometer was equipped with an inlet that was heated to 300°C to remove non-refractory
       components from the aerosol phase. The $M_{BC}$ values measured by COSMOS and those measured by the
       University of Tokyo SP2 agreed to within about 10% at Ny-Ålesund (Ohata et al., 2019). Because
COSMOS data were not obtained at these two sites in 2008 and 2010, data obtained by particle
       absorption soot photometers (PSAP; Radiance Research, Seattle, WA, USA, Ogren et al., 2017) were
       used after scaling their $M_{BC}$ values to agree with COSMOS values by using a scaling factor derived
       from side-by-side measurements (Sinha et al., 2017; Ohata et al., 2020). Daily averaged data were used
       in this study.

**3 Numerical model simulations**

       The two climate-aerosol models, the Community Atmosphere Model version 5 with the Aerosol Two-
       dimensional bin module for foRmation and Aging Simulation (CAM5-ATRAS), and the Meteorological
       Research Institute Earth System Model version 2.0 (MRI-ESM2), were used in this study.

       The CAM5-ATRAS model calculates major chemical and aerosol processes in the atmosphere (Matsui,
2017; Matsui and Mahowald, 2017; Matsui et al., 2018). The model considers seven aerosol species
       (sulfate, nitrate, ammonium, dust, sea salt, organic aerosol, and BC) with a two-dimensional sectional
       representation consisting of 12 bins of particle sizes from 1 to 10,000 nm dry diameter and 8 bins of BC
       mixing states for fine particles (40 to 1250 nm). Simulations were performed with a horizontal
       resolution of 1.9° × 2.5° latitude/longitude and 30 vertical layers. Meteorological nudging used data for
temperature and horizontal wind in the free troposphere (<800 hPa) from the Modern-Era Retrospective
       analysis for Research and Applications version 2 (MERRA2) dataset (Gelaro et al., 2017). We used
       monthly AN emission from the CMIP6 dataset (Hoesly et al., 2018) and daily BB emission from the
       Global Fire Emissions Database version 4.1 (GFED, van der Werf et al., 2017). Because CMIP6
       emissions are available only for years 1750–2015, AN emission in 2015 were used for calculations for
the year 2018. BC from AN (fossil fuel + biofuel) and BB sources were tracked separately to calculate
       their individual contributions to total BC.

       MRI-ESM2 is an atmosphere–ocean coupled model that integrates interactive models for aerosol and
       atmospheric chemistry (Yukimoto et al., 2019; Kawai et al., 2019; Oshima et al., 2020) that participated
       in the CMIP6 (Eyring et al., 2016). The model configurations (i.e., use of the atmospheric general
circulation model and aerosol model) and calculation methods are the same as those in Adachi et al.
       (2021). The model employs TL159 horizontal resolution (approximately 120 km) and 80 vertical layers
       from the surface to 0.01 hPa (the model top) in a hybrid sigma-pressure coordinate system. The model
       simulation was performed from January 2008 to December 2018 after a 1-year spin-up run with the



prescribed sea surface temperature and sea ice data (Ishii et al., 2005). In the simulations, horizontal

wind fields were nudged toward 6-hourly Japanese 55-year Reanalysis data (Kobayashi et al., 2015).

Simulations used monthly AN emission from the MACCity emission dataset (Lamarque et al., 2010)

and daily BB emission from the Global Fire Assimilation System dataset (GFAS, Kaiser et al., 2012).

We also performed a model simulation that omitted the BB emission of BC to estimate the separate

contributions of AN and BB sources to total BC concentrations.

**4 Year-to-year variation in BC and influences from BB**

**4.1 Vertical profile and column amounts of BC mass concentration**

Fig. 2a shows a vertical profile of $M_{BC}$ using all of the one-minute data obtained during the

PAMARCMiP aircraft experiment. Median values in individual altitude ranges are also shown and

listed in Table 2. Median $M_{BC}$ values were between 7 and 18 ng m$^{-3}$ at altitudes below 4.5 km. At

altitudes around 5 km, enhancements of $M_{BC}$ up to 250 ng m$^{-3}$ (one-minute data) were observed. As

described in Sect. 5, these high $M_{BC}$ values were likely due to influences from BB BC emissions.

In Fig. 2b, the vertical profile of median $M_{BC}$ values obtained during the PAMARCMiP experiment is

compared with those obtained during the ARCTAS, HIPPO, and NETCARE aircraft experiments made

over the Arctic in spring (Table 3). Median values were calculated only from data obtained at latitudes

north of 66.5°N, the latitude of the Arctic Circle. This latitude generally corresponded to locations of a

boundary of the polar dome, in which Arctic air mass tends to be confined, as estimated for the

NETCARE 2014 and 2015 experiments (Bozem et al., 2019). The PAMARCMiP $M_{BC}$ values were

generally similar to those obtained during the HIPPO experiment, whereas systematically higher values

were observed during the ARCTAS and NETCARE experiments. Although each aircraft measurement

was made over a limited area and time duration, they document significant year-to-year variations in

$M_{BC}$ values in the Arctic spring. In the following sections, BC mass concentrations obtained during

these four aircraft experiments are examined.

Using the four aircraft datasets, BC column amounts at altitudes between 0 and 5 km ($COL_{BC}$, in units

of μg m$^{-2}$) were calculated by vertically integrating median $M_{BC}$ values in each altitude obtained at

latitudes north of 66.5°N (Table 4). Although data were obtained at altitudes higher than 5 km during

the ARCTAS and HIPPO experiments, they were not used. Fig. 3 compares $COL_{BC}$ values among the

four experiments. In accordance with the vertical $M_{BC}$ profiles (Fig. 2b), $COL_{BC}$ values are similar

between the PAMARCMiP and HIPPO experiments (within a factor of 1.4). The $COL_{BC}$ values for the

NETCARE and ARCTAS experiments are about a factor of 2.7 and 4.2, respectively, higher than the

PAMARCMiP value (Table 4).



A part of the differences in the observed BC levels could be due to the different locations where the measurements were made. As shown in Fig. 1a, the PAMARCMiP and NETCARE aircraft experiments were conducted mainly at latitudes around 80°N, while ARCTAS and HIPPO data obtained at latitudes between 66.5 and 90°N were used in this study. In Fig. 3 and Table 4, $COL_{BC}$ values calculated using
only data obtained at latitudes north of 80°N are also shown, although the amount of data for HIPPO is limited. The ARCTAS $COL_{BC}$ value for >80°N is lower only by 5% than the value for >66.5°N, and HIPPO $COL_{BC}$ value for >80°N is even higher than the value for >66.5°N. Consequently, systematic differences in $M_{BC}$ and $COL_{BC}$ values were likely not due to the different observation locations. This conclusion is also supported by the numerical model simulations described in Sect. 4.3, where $COL_{BC}$
values from the same area (entire Arctic region) are compared for the four time periods.

## 4.2 Biomass burning fire counts

In Fig. 3, the daily averaged numbers of fire counts (counts day$^{-1}$, indicator of BB activities) detected by the MODIS satellite (MCD14DL products, https://earthdata.nasa.gov/active-fire-data) are shown for the four aircraft experiments. In this figure, we show fire counts at latitudes north of 50°N for the time
period between 14 days before the first day of the aircraft experiment and the last day of the experiment. The averaged fire counts for HIPPO and PAMARCMiP are similar, whereas those for NETCARE and ARCTAS are, respectively, factors of around 5 and 14 higher than that for PAMARCMiP (Fig. 3 and Table 4). Similar tendencies can be also seen in averaged GFAS and GFED BB BC emissions (Fig. 3) that were created using the fire counts and were used for numerical simulations in this study (section 3).
Consequently, the relative changes in $COL_{BC}$ were generally consistent with those in BB activities. These results suggest that BC emissions from BB likely contributed to the increased $COL_{BC}$ during the NETCARE and ARCTAS experiments, at least to some extent. In other words, year-to-year variations of tropospheric BC amounts in the Arctic in spring were primarily affected by transport of BB BC, at least during these four periods. As described below in Sect. 4.3, results from numerical model
simulations indicate that the year-to-year variation of AN $COL_{BC}$ (>66.5°N) was smaller than 30–40%, thus supporting that year-to-year variation in $COL_{BC}$ was mostly due to BB BC.

Fig. 4 shows a map of averaged fire counts for the four experiment time periods. Fire counts were generally high at latitudes 45–60°N and longitudes around 30–50°E and 100–130°E (western and eastern Eurasia, respectively). Year-to-year variations in BB activities in these areas contributed to
cause the variations in average fire counts shown in Fig. 3.

As described above, we chose a time period of 14 days before the start date of an aircraft experiment for the calculation of the averaged fire counts in consideration of the possible lifetime of BC in the Arctic and the transport time of BC from its sources. The time period of 14 days is somewhat longer than the estimated globally and annually averaged BB-originated BC lifetime of 6–7 days (Koch and Hansen,



2005; Park et al., 2005; Evangeliou et al., 2016). Even when this criterion was changed to 0, 7, and 21 days before the first day of the experiment, the relative changes in averaged fire counts shown in Fig. 3 were not strongly affected. A time series of fire counts in individual latitude ranges is presented in Fig. C1 in Appendix C. As seen in this figure, fire counts at latitudes > 40°N and > 50°N started to increase in mid- to late March during each of the four years examined in this study. Because time periods of 7 or

14 days generally captured this onset of fire activities, the year-to-year variations of fire counts were generally similar irrespective of the chosen time period. Notably, time series of the fire counts at latitudes > 40°N and > 50°N were generally similar (Fig. C1) and year-to-year variations were also similar when either of these criteria was adopted. Furthermore, the fire counts at latitudes > 60°N were negligible in all the cases and the correspondence to their year-to-year variation was less significant.

Consequently, transport of air influenced by BB at latitudes between 45°N and 60°N is likely responsible to the increased BC level in the Arctic spring.

As described in Sect. 2, during the PAMARCMiP experiment, air parcels sampled at altitudes below 1, 3, and 5 km had generally originated from north of 70°N, 60°N, and 50°N, respectively (Fig. B1). Most of the air parcels sampled at altitudes above 3 km had not passed over high BB areas and had

maintained their altitudes, thus, they were likely not significantly influenced by BB BC emissions. Consequently, air parcels sampled during the PAMARCMiP experiment were likely not influenced by recent BC emissions, except for the plumes observed at altitudes around 5 km (Sect. 5).

### 4.3 Evaluation of BB BC using numerical model simulations

In Fig. 5, the observed $COL_{BC}$ (>66.5°N) values are compared with numerical model simulations from

CAM5-ATRAS and MRI-ESM2, described in Sect. 3. The model-calculated $COL_{BC}$ values were derived in three different ways by calculating median or average values for individual altitude ranges: (A) median values along the flight tracks, (B) area-weighted averages within the latitudes and longitudes and the time periods of the aircraft experiments shown in Table 3, and (C) area-weighted averages within the entire region at latitudes north of 66.5°N for the time period of an aircraft

experiment (Table 5). Because medians were calculated to derive observed $COL_{BC}$ values (Fig. 3 and Table 4), $COL_{BC}$(A) values can be directly compared with observations. Area-weighted averages were calculated to derive $COL_{BC}$(B) and $COL_{BC}$(C) values, because medians cannot be calculated for gridded model values that have different areal weights depending on the latitudes. The reason for calculating the statistics in three ways is that temporal and spatial variations of BC may not be accurately reproduced in

the numerical model simulations and it may make more sense to compare statistics over a wider spatial area. Furthermore, differences between $COL_{BC}$(B) and $COL_{BC}$(C) values show possible differences in $M_{BC}$ values among the different observational areas of the four experiments. Fig. 5 and Table 5 show that $COL_{BC}$(B) values are generally lower than $COL_{BC}$(C). $COL_{BC}$(B)/$COL_{BC}$(C) ratios were found to





be higher for HIPPO and PAMARCMiP (0.79–0.89) than those of ARCTAS and NETCARE (0.54–

0.66), although the observed $COL_{BC}$ values were higher for the latter two experiments as compared with the former ones. These results suggest that the observed higher $COL_{BC}$ values during the ARCTAS and NETCARE experiments likely not due to the different observational areas but mostly due to year-to-year variations in $M_{BC}$ over the entire Arctic.

Fig. 5 shows that both models reproduced the observed $COL_{BC}$ values well for PAMARCMiP and

HIPPO, whereas they significantly underestimated the observed $COL_{BC}$ values for ARCTAS (by factors of 3.7–5.8) and NETCARE (by factors of 3.3–5.0), when influences from BB BC emissions are considered to have been high. Consequently, the significant underestimations in $COL_{BC}$ could be mostly due to those in BB BC.

In Fig. 5, model-calculated contributions from BB and AN emissions are shown separately. In general,

the year-to-year variation of the model-calculated AN values was small (at most 30 or 38% for the two model results for >66.5°N, $COL_{BC}$(C)), while that of BB values was much higher (by at most a factor of 15 or 28). When model simulations reproduced the observations well, namely those from the HIPPO and PAMARCMiP experiments, most of the values of the $COL_{BC}$(A) originated from AN (98% and 82–94% for CAM5-ATRAS and MRI-ESM2, respectively). Consequently, the AN contributions are

considered to be generally well reproduced. However, although the BB BC values were higher in the ARCTAS and NETCARE cases, the total (AN+BB) $COL_{BC}$(A) values were still significantly lower than the observations. Reasonable agreement was found only when the calculated BB BC contributions were multiplied by a factor of 3 or more (C′ in Fig. 5). As described in Sect. 3, CAM5-ATRAS and MRI-ESM2 utilized GFED and GFAS BB BC emissions, respectively. Consequently, these results

suggest that BB BC emissions could be largely underestimated or the removal of BC from BB emissions was overestimated, although the level of these uncertainties cannot be evaluated in this study.

### 4.4 Correspondence to surface BC measurements

Fig. 2b shows that $M_{BC}$ values near the surface were higher for ARCTAS and NETCARE as compared with those of PAMARCMiP and HIPPO, although larger increases were observed in the free

troposphere, namely above 1 km altitude. How much of the year-to-year variations in $M_{BC}$ observed in the free troposphere was observed at the ground surface? To evaluate the level of the variation, we examined surface BC data obtained in Ny-Ålesund and Barrow (Sect. 2.3). Fig. 6 shows the medians of daily averaged $M_{BC}$ values during the individual aircraft experiments observed at the surface in Ny-Ålesund and Barrow. Because the number of days for an experiment was relatively small (2 and 7 days

for HIPPO and NETCARE, respectively), medians were also calculated for the 31-day period for which the median date was chosen to be the median date of the experiment (Fig. 6 and Table 6). The corresponding average fire counts (31 + 14 days) are also shown in Fig. 6 and Table 6.



Previous long time period studies showed that $M_{BC}$ values in Barrow were about a factor of 1.4 higher than in Ny-Ålesund in March–April (Sinha et al., 2017). This tendency is also recognized in the
ARCTAS and PAMARCMiP periods shown in Fig. 6, although it is not clearly seen in the other two periods. When the 31-day median values at these two sites were examined, $M_{BC}$ values for the ARCTAS year were higher by factors of 2.5 – 3.2 in comparison to the PAMARCMiP period. This relative change was somewhat smaller than for $COL_{BC}$ values (by factors of 4.2 and 3.6 for >66.5°N and 80°N, respectively), however we can still conclude that surface BC measurements can capture the
year-to-year variations of atmospheric $COL_{BC}$ caused by varying BB activities. The smaller year-to-year variation observed at the surface was likely due to a strong temperature inversion that inhibited vertical mixing (Brock et al., 2011).

**5 Case study of BB BC transport**

**5.1 Origin of air parcels with enhanced $M_{BC}$**

As described in Sect. 4, the atmospheric BC level during the PAMARCMiP experiment was least affected by recent biomass burning at altitudes below 4.5 km. Considering the negligible levels of BB activities before the experiment (Fig. C1) and the very small BB contributions predicted by numerical models (Fig. 5), most BC particles observed during PAMARCMiP were supposed to be of anthropogenic origin. This is also supported by observational results of Yoshida et al. (2020) and
Adachi et al. (2021), who found tight correlations between BC and anthropogenic iron oxide particles during PAMARCMiP. On the other hand, plumes with enhanced $M_{BC}$ values up to 250 ng m$^{-3}$ were observed at altitudes around 5 km, as shown by the red data points in Fig. 2a. In this section, we propose that these enhanced $M_{BC}$ values were likely due to influences from BB emissions. We then examine possible differences in microphysical properties between these BC particles and other BC
particles that were least affected by BB.

The plumes with enhanced $M_{BC}$ values were observed at similar locations and altitudes on 2, 3, and 4 April. In total, 29 one-minute data with $M_{BC} > 50$ ng m$^{-3}$ were obtained. Although the horizontal wind direction was different on these three days, wind speeds of 2.5 to 6.5 m s$^{-1}$ suggest that the plumes could extend over 1000 km. Because the research aircraft could not fly at altitudes above 5.2 km, we
could not observe the vertical extent of these enhancements. However, the elastic air-borne mobile aerosol Lidar (AMALi) installed on board of the research aircraft observed a well-defined aerosol layer at altitudes between 5.15 and 6.8 km on 2 April (Nakoudi et al., 2020), suggesting that the plumes with enhanced $M_{BC}$ values could extend over this vertical range. A photo taken from the research aircraft POLAR 5 on 3 April is also shown in Appendix D (Fig. D1), where a haze layer with reddish color,
which may correspond to the plumes, was captured.



When $M_{BC}$ enhancements were observed, increased aerosol potassium concentration was also observed, suggesting that these plumes had likely been affected by BB emissions (Adachi et al., 2021). Enhancements in CO and $CO_2$ concentrations were also observed in these plumes. The ratio of their increases ($\Delta CO/\Delta CO_2$), derived from a slope of a least squares fit to the scatter plots of CO and $CO_2$ concentration data during the plume periods, was 12.3 ppbv ppmv$^{-1}$, suggesting that these plumes were likely due to incomplete combustion. This result is also in accordance with influences from BB. In fact, this ratio is similar to the $\Delta CO/\Delta CO_2$ ratio of 15±5 ppbv ppmv$^{-1}$ observed in air affected by Asian BB observed during the ARCTAS experiment (Kondo et al., 2011a).

Fig. 7a shows 8-day backward trajectories of air with enhanced $M_{BC}$ values. These air parcels originated from eastern Eurasia where active BB took place when air parcels were located. They were likely uplifted upon the passage of a cold frontal system over the BB area. MRI-ESM2 model simulations generally reproduced these transport processes. Fig. 7b shows that air with enhanced $M_{BC}$ values was transported on isentropic surfaces up to 450 hPa level along 135°E. The air then spread in horizontal direction at the 500 hPa level. Enhanced $M_{BC}$ values were likely observed within this air (Fig. 7c). The transport pathway of this BB plume is also discussed by Adachi et al. (2021).

The GFED mass emission ratio of BC to CO from BB sources in eastern Eurasia shown in Fig. 7a (40.7°N–52.1°N and 110°E–125°E during 25–27 March, eight days before the observations) was 5.54 × 10$^{-3}$ ng m$^{-2}$ s$^{-1}$ (ng m$^{-2}$ s$^{-1}$)$^{-1}$, which corresponds to $[\Delta BC/\Delta CO]_{source}$ = 6.92 ng m$^{-3}$ ppbv$^{-1}$ at STP. Within the plumes, the slope of $[\Delta BC/\Delta CO]_{air}$ = 4.00 ng m$^{-3}$ ppbv$^{-1}$ at STP was observed. Because CO molecules are not removed by precipitation while BC particles are, the transport efficiency of BC can be estimated to be $TE_{BC}$ = $[\Delta BC/\Delta CO]_{air}$/ $[\Delta BC/\Delta CO]_{source}$ = 0.58, indicating that 42% of the BC emitted from BB had been removed during the transport. During the ARCTAS experiment, $[\Delta BC/\Delta CO]_{source}$ was estimated to be 7.6 ng m$^{-3}$ ppbv$^{-1}$ at STP within Russian BB plumes, which were likely the least affected by wet deposition (Matsui et al., 2011); this estimate is in accordance with the estimate in this study (6.92 ng m$^{-3}$ ppbv$^{-1}$) using the GFED data. The observed $[\Delta BC/\Delta CO]_{air}$ of about 6.3 ng m$^{-3}$ ppbv$^{-1}$ at STP and the resulting $TE_{BC}$ of 0.83 obtained during the ARCTAS experiment for the Russian BB air were slightly higher than the values obtained in this study (4.0 ng m$^{-3}$ ppbv$^{-1}$ at STP and 0.58). The lower PAMARCMiP $TE_{BC}$ values are in accordance with the fact that the accumulated precipitation along the backward trajectories (e.g., Oshima et al., 2012; Raut et al., 2017) was about two times higher than that of the ARCTAS experiments (i.e., 19.7 mm for PAPAMACMiP and 8.5 mm for ARCTAS), suggesting that BC had been more efficiently removed by precipitation during PAMARCMiP.



### 5.2 Microphysical features of BB BC

Fig. 8a shows a vertical profile of the mass averaged diameter of BC particles ($D_{BC}$), which was defined as

$$D_{BC} = \left( \frac{6M_{BC}}{\pi \rho_{BC} N_{BC}} \right)^{\frac{1}{3}}, \qquad (1)$$

where $\rho_{BC}$ is the void-free density of BC (1.8 g cm$^{-3}$) and $N_{BC}$ is the number concentrations of BC for the $D_m$ range of 70–850 nm. The median $D_{BC}$ values in individual altitude ranges are also shown. Enhanced $M_{BC}$ value data are shown in red color. The median $D_{BC}$ values decreased with increasing altitude; they were about 167 nm near the ground surface and 134 nm at 4.5 km (Table 2). The normalized mass size distribution of BC shows a gradual shift of the mode diameter to a smaller size at higher altitudes (Fig. 9), which is consistent with that observed during NETCARE experiment (Schulz et al., 2019). The values of mass median diameter (MMD) and geometric standard deviation ($\sigma_g$) for the lognormal fitted size distributions are also shown in Fig. 9. A previous study made over East Asia showed that the $D_{BC}$ values observed in the free troposphere was generally smaller than those in the planetary boundary layer influenced by anthropogenic emissions, likely due to wet removal during upward transport (Moteki et al., 2012). In fact, when the wet removal fraction was higher (TE$_{BC}$ was lower), the $D_{BC}$ was smaller in the free troposphere. Because larger diameter BC particles generally have greater amounts of coating material and hence, have higher cloud condensation nuclei (CCN) activities, they are selectively removed by precipitation (Moteki et al., 2012; Ohata et al., 2016; Moteki et al., 2019). When air parcels are uplifted to higher altitudes, they potentially experience higher supersaturation in clouds, and aerosols with lower CCN activities (smaller diameter of BC core) can remain. These mechanisms likely resulted in the observed smaller $D_{BC}$ values at higher altitudes during the PAMARCMiP experiment.

The median value of the $D_{BC}$ in the enhanced $M_{BC}$ air was 145 nm (Table 2), which is only slightly higher than that of other BC particles sampled at this altitude range. Various factors could cause the observed differences. The BC particles could have already been larger upon BB emission than those from anthropogenic sources or they could have been removed less efficiently. However, considering the relatively large change in $D_{BC}$ with altitude (Figs. 8a and 9), the similarity in $D_{BC}$ between recent BB plumes and aged AN air is remarkable.

Fig. 8b shows the vertical profile of the median shell-to-core diameter ratios for $D_m = 180$–192 nm. It is remarkable that the shell-to-core ratios change so little with altitude. The median values are between 1.40 and 1.48 over the entire altitude range (Table 2). This tendency and similar values were also observed during the NETCARE experiment (Kodros et al., 2018), suggesting that the observed features are common in the Arctic spring. Considering the possible greater influences of wet removal at higher altitudes, the small change in the shell-to-core ratios with altitude was not easily interpreted.



The shell-to-core diameter ratios of the enhanced $M_{BC}$ data were only slightly higher than for other data (Table 2). These air parcels were likely affected by recent (within a week) wet removal and, therefore, one may expect a smaller coating thickness because BC particles with thicker coating were likely
removed due to higher CCN activities. Weaker updraft speed (lower supersaturation) or less hydrophilic coating compositions could be possible explanations. Cloud-precipitation processes during transport such as the aqueous-phase reactions or evaporation of precipitating particles after accretion might also increase the coating thickness of BC. The similarity in the shell-to-core diameter ratios between recent BB plumes and aged AN air provide useful constraints to validate numerical model simulations.

**6 Summary**

The PAMARCMIP aircraft-based experiment measured vertical profiles of $M_{BC}$ at altitudes up to 5 km in spring 2018. Median $M_{BC}$ values in individual altitude ranges were 7–18 ng m$^{-3}$ at altitudes below 4.5 km. These concentrations were systematically lower than those observed during ARCTAS in 2008 and NETCARE in 2015, whereas they were similar to those observed during HIPPO in 2010. Vertically
integrated BC mass concentrations for altitudes below 5 km in the Arctic (>66.5°N, $COL_{BC}$) observed during the ARCTAS and NETCARE experiments were higher than the PAMARCMiP value by factors of 4.2 and 2.7, respectively. When only data obtained at >80°N were compared, the same tendency was observed: the ARCTAS and NETCARE $COL_{BC}$ values were higher by factors of 3.6 and 2.3, respectively. The year-to-year variation of $COL_{BC}$ values generally corresponded to that of MODIS-
derived BB fire counts in northern high latitudes (45–60°N) in western and eastern Eurasia (around 30–50°E and 100–130°E, respectively). Furthermore, two numerical model simulations estimated that the year-to-year variation of contributions from AN emission was lower than 30–40%. These results suggest that the year-to-year variation of BB activities likely affected the BC amounts in the Arctic troposphere in spring, at least in the years examined in this study.

Numerical model simulations generally well reproduced the observed $COL_{BC}$ values for PAMARCMiP and HIPPO (within factors of 2) when BB activities were low, but underestimated the values for ARCTAS and NETCARE (by factors of 3.3 to 5.8) when BB activities were high. Because AN contribution mostly accounts for PAMARCMiP and HIPPO $COL_{BC}$ (82–98%), the good agreements between the observations and model simulations for these two experiments suggest that AN contribution
was generally well reproduced. However, the significant underestimations for ARCTAS and NETCARE suggest that the BB contributions were underestimated. For these two cases, reasonable agreement was found only when the calculated BB BC contributions were multiplied by a factor of 3 or more. Consequently, these results suggest that BB BC emissions could be greatly underestimated or the removal of BC from BB emissions was overestimated, although the level of these uncertainties cannot
be evaluated in this study.





The year-to-year variation of $M_{BC}$ values obtained at the surface at the high Arctic sites Ny-Ålesund and Barrow during the periods corresponding to the four aircraft experiments was also examined. The 31-day median $M_{BC}$ values for the ARCTAS year were higher by factors of 2.4–3.2 than for the PAMARCMiP period. Although this relative change in $M_{BC}$ values was somewhat smaller than that in the $COL_{BC}$ values (by factors of 4.2 and 3.6 for >66.5°N and 80°N, respectively), likely due to the limited vertical mixing, we can still conclude that ground surface BC measurements can capture the year-to-year variations caused by BB activities.

Because BC particles observed at altitudes below 4.5 km during the PAMARCMiP experiment were least affected by recent BB and mostly originated from AN source, the microphysical properties of BC particles were studied. The mass averaged diameter of BC particles ($D_{BC}$) decreased with increasing altitude, from 167 nm near the ground surface to 134 nm at 4.5 km. This tendency is in accordance with the selective removal of BC containing particles with higher CCN activities. The median shell-to-core diameter ratios for $D_m$ = 180–192 nm changed little with altitude and were between 1.40 and 1.48 throughout the entire altitude range. Interestingly, $D_{BC}$ and the shell-to-core diameter ratio of BC in the BB plumes observed during the PAMARCMiP experiment were generally not very different from those in other air sampled, which were considered to be mostly aged AN BC.

Direct radiative forcing of BC in the Arctic is considered to be highest in spring when $M_{BC}$ is the largest and incoming solar radiation is increasing. BC in spring is also important because slight changes in the time of snow/ice melt can influence the radiation budget in the Arctic. The observations presented in this study provide useful bases to improve and evaluate numerical model simulations that assess the BC radiative effect in the Arctic spring.

**Appendix A. Comparison between the University of Tokyo and NOAA SP2 measurements**

$M_{BC}$ measurements made with the University of Tokyo SP2 during ARCTAS (Sahu et al., 2012) were compared with NOAA SP2 measurements (NOAA-ARCPAC experiment, Spackman et al., 2010). Two measurements were made onboard the NASA DC-8 and NOAA WP-3D, respectively. Fig. A1a shows that the two aircraft flew very close-track at altitudes between 0.3 and 5.6 km for about 70 min on 12 April 2008. This figure also shows that time series of one-minute $M_{BC}$ data agreed well between the two measurements. Fig. A1b shows a scatter plot between these two one-minute data. They were highly correlated ($R^2$ = 0.92) with a slope of 0.80. The systematically higher U. Tokyo values might partly be attributed to a wider size range of U. Tokyo BC measurements and a difference in corrections for undetected BC between the measurements as described below.

During the ARCTAS campaign, the U. Tokyo SP2 measured BC particles for the $D_m$ range of 80–860 nm. Although the estimated mass fraction of the undetected BC was small (4% for 50-80 nm), the $M_{BC}$ for the $D_m$ range of 50–900 nm was obtained by integrating the single-mode lognormal fit function of

the measured size distribution (Sahu et al., 2012). On the other hand, the measured $D_m$ range of the NOAA SP2 during the NOAA-ARCPAC experiment was 90–600 nm. The $M_{BC}$ was estimated by multiplying 1.1 to the integrated mass of BC particles, considering the undetected mass fractions of BC outside the detection range (Spackman et al., 2010, Schwarz et al., 2010). These differences in measurements and data reductions could result in some systematic differences between the two $M_{BC}$

datasets.

Because $M_{BC}$ measurements were consistently and traceably made by the individual groups, the systematic differences in absolute $M_{BC}$ values obtained by them are expected to be smaller than 20%.

**Appendix B. Back trajectories of sampled air**

Six-day kinematic back trajectories of air parcels measured onboard the aircraft were calculated every

minute based on the method described by Tomikawa and Sato (2005). For calculating the trajectories, 6-hourly meteorological data from the National Centers for Environmental Prediction (NCEP) Final (FNL) operational global analysis were used. As shown in Fig. B1, air parcels sampled at altitudes below 1 km and between 1 and 3 km had originated mostly from north of 70°N and 60°N, respectively. Air parcels sampled at altitudes between 3 and 5 km had occasionally originated at latitudes as low as

50°N, in central and eastern Eurasia and north-western Canada.

**Appendix C. Time series of fire counts**

Time series of daily averaged number of fire counts (counts day$^{-1}$) detected by MODIS satellite (MCD14DL products, https://earthdata.nasa.gov/active-fire-data) are shown in Fig. C1 for the four aircraft experiments at latitudes north of 40°N, 50°N, and 60°N. The first day of individual aircraft

experiments (>66.5°N) are shown. Fire counts at latitudes > 40°N and > 50°N started to increase in mid-to late March in all four of the years examined in this study.

**Appendix D. Photo taken from POLAR 5**

A photo taken from the research aircraft POLAR 5 on 3 April is shown in Fig. D1. A haze layer with reddish color, which may correspond to the plumes, was captured. Although not so apparent as this

photo, a haze layer with dark color was also sighted on April 2 and 4, when the plumes with enhanced $M_{BC}$ values were observed.

**Data availability**

The observational data set used in this publication will be available online (https://ads.nipr.ac.jp).





**Author contributions**

SO and MK designed the study and wrote the paper. SO, MK, AY, NM, KA, OE, HB, MZ, and ABH contributed to the PAMARCMiP experiments and data analysis of SP2, TEM, and CO and $CO_2$ gas analysers. NO and HM conduced numerical model simulations using MRI-ESM2 and CAM5-ATRAS, respectively.

**Competing interests**

The authors declare that they have no conflicts of interest.

**Acknowledgements**

We are indebted to all the PAMARCMiP 2018 participants for their cooperation and support. The authors also acknowledge the Alfred Wegener Institute (AWI) for both the support to conduct the PAMARCMiP 2018 campaign and the use of the Polar 5 research aircraft and the skill and safety
exemplified by the pilots and flight staff. We thank the biogeochemistry department of MPIC for providing the CO instrument and Dieter Scharffe for his support during the preparation phase of the campaign. We thank Yutaka Kondo, Bryan Thomas, Peter Detwiler, and Ross Peterson, and Norwegian Polar Institute for supporting the ground-based COSMOS measurements at Barrow and Ny-Ålesund. We also thank J. Ryan Spackman and Joshua P. Schwarz for making comparison measurements of BC
during the ARCTAS and ARCPAC experiments. This research was performed by the Environment Research and Technology Development Fund (JPMEERF20205001, JPMEERF20202003, JPMEERF20172003, and JPMEERF20165005) of the Environmental Restoration and Conservation Agency of Japan, the Global Environmental Research Coordination System of the Ministry of the Environment of Japan (MLIT1753). This study was also supported by the Ministry of Education,
Culture, Sports, Science, and Technology and the Japan Society for the Promotion of Science (MEXT/JSPS) KAKENHI Grant Numbers JP17H04709, JP18H03363, JP18H05292 and JP20H00638, and the Arctic Challenge for Sustainability (ArCS) project (JPMXD1300000000) and ArCS II (JPMXD1420318865) project of the MEXT of Japan. MK thanks Ikuyo Oshima and Takumi Iwata for their contributions to the data analyses. We acknowledge the use of the MODIS/Aqua+Terra Thermal
Anomalies/Fire locations 1km V006 NRT (Vector data) distributed by LANCE FIRMS NASA Near Real-Time and MCD14DL MODIS Active Fire Detections (TXT format), which are available online (https://earthdata.nasa.gov/active-fire-data). We also gratefully acknowledge the funding by the Deutsche Forschungsgemeinschaft (DFG, German Research Foundation) – project ID 268020496 – TRR 172, within the Transregional Collaborative Research Center "ArctiC Amplification: Climate
Relevant Atmospheric and SurfaCe Processes, and Feedback Mechanisms (AC)[3]".



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





**Figures**

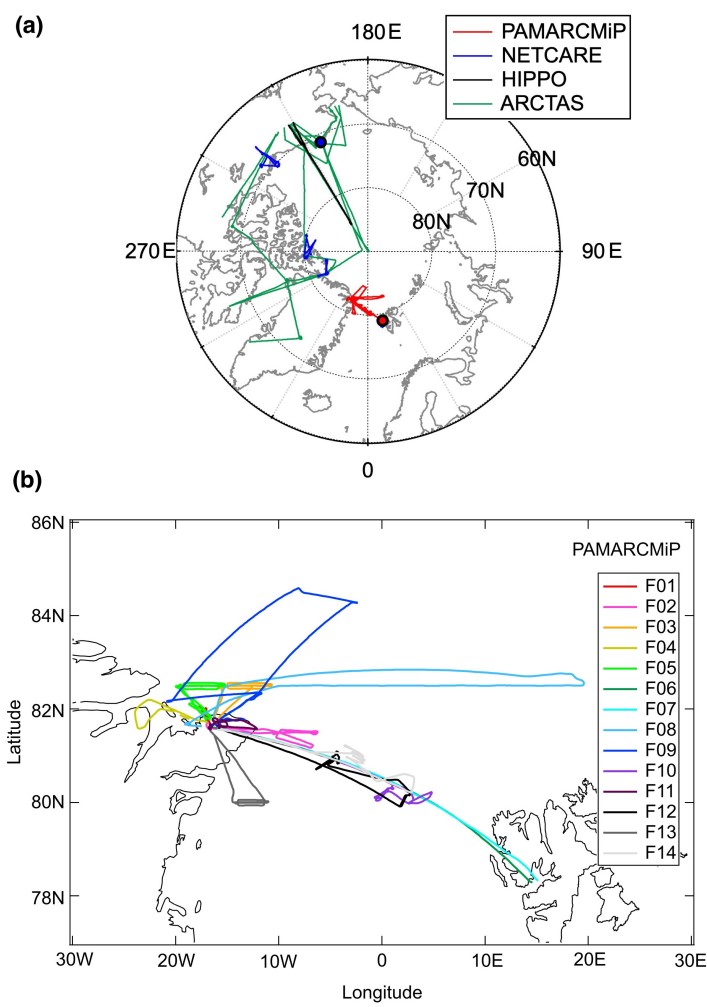

**Figure 1.** (a) Map of the flight tracks of the PAMARCMiP aircraft-based field experiment and the other three experiments made over the Arctic in spring. In this study, only data obtained at latitudes north of 66.5°N were used and only the flight tracks for them are shown. The locations of the high Arctic sites Ny-Ålesund in Svalbard (78.9°N, 11.9°E) and Barrow in Alaska (71.3°N, 156.6°W) are indicated by the red and blue circles, respectively. (b) Map of the detailed flight tracks of the PAMARCMiP experiment.








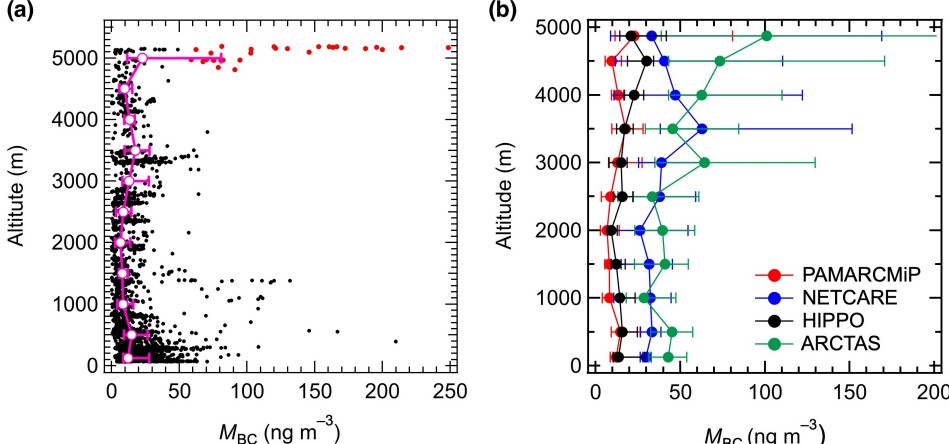

**Figure 2.** (a) Vertical profile of the BC mass concentration ($M_{BC}$) observed during the PAMARCMiP aircraft-based field experiment. Closed circles denote one-minute data. Median values and 25–75% ranges are also shown. Data points shown in red (enhanced $M_{BC}$ data points) were likely influenced by biomass burning emissions and are discussed in Sect. 5. (b) Vertical profile of median $M_{BC}$ values observed during the aircraft-based experiments made over the Arctic (>66.5°N) in spring (Table 3).





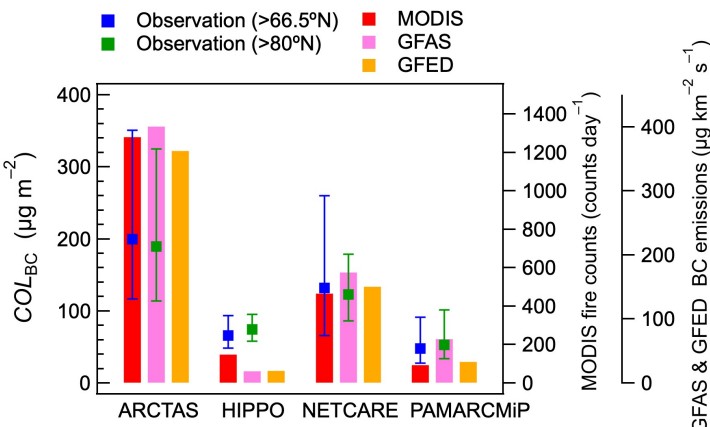

**Figure 3.** Column amounts of BC mass concentration at altitudes between 0 and 5 km ($COL_{BC}$) at latitudes north of 66.5°N (blue squares) and 80°N (green squares). MODIS-derived fire counts and biomass burning BC emissions compiled in GFAS and GFED datasets are also shown. The fire counts and BC emissions are those at latitudes north of 50°N for the time period between 14 days before the first day of an aircraft experiment and the last day of the experiment.


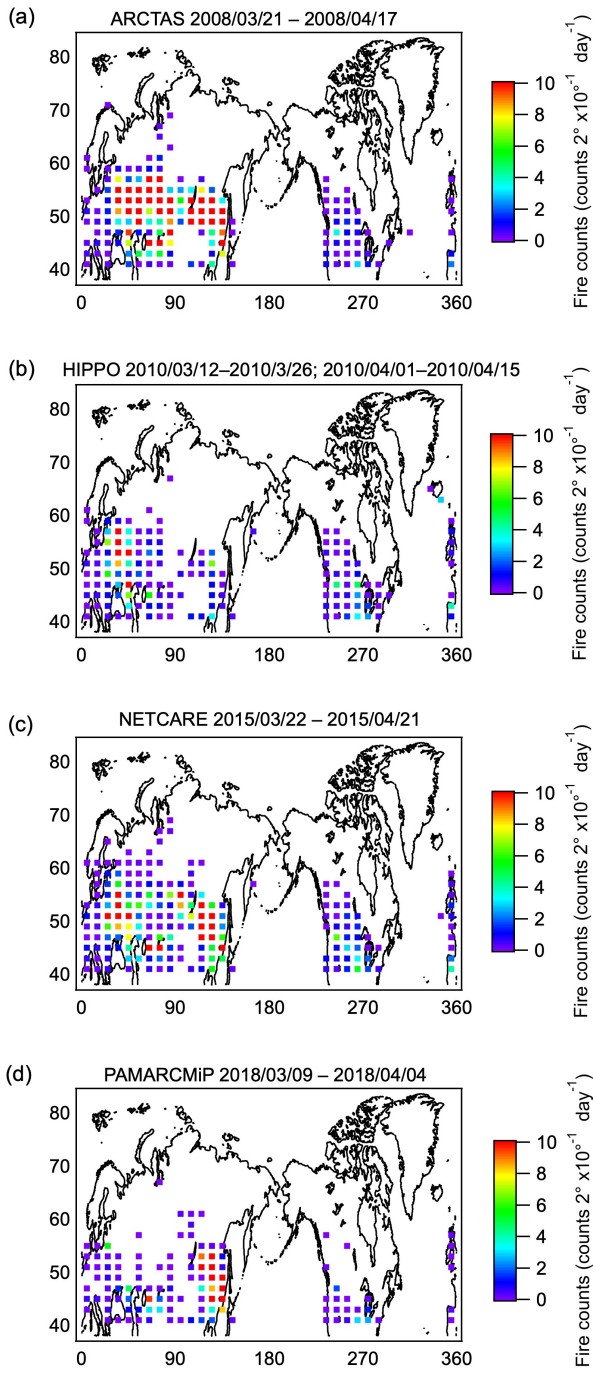

**Figure 4.** Map of MODIS-derived fire counts. Averaged values are shown for the time period between 14 days before the first day of an aircraft-based experiment and the last day of the experiment.




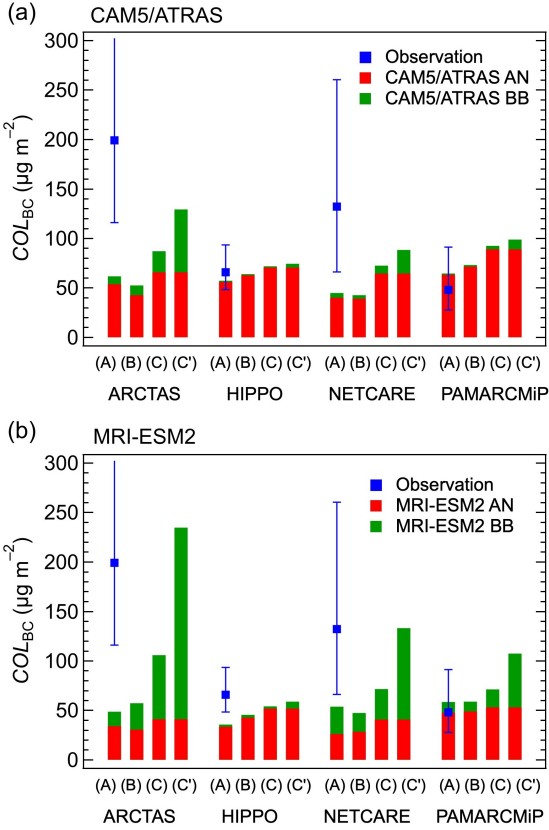

**Figure 5.** Observed column amounts of BC mass concentration at altitudes between 0 and 5 km and latitudes >66.5°N ($COL_{BC}$, blue squares) and model calculated $COL_{BC}$ values (vertical bars). For calculated values, contributions from anthropogenic (AN) and biomass burning (BB) emissions are shown by red and green colors, respectively. The model-calculated $COL_{BC}$ values were derived in three different ways by calculating median or average values for individual altitude ranges: (A) Median values along the flight tracks. (B) Area-weighted averages within the areas and time periods of the individual aircraft-based experiments shown in Table 3. (C) Area-weighted averages within the entire region at latitudes north of 66.5°N for the time periods of the individual aircraft experiments. (C′) Same as (C) but with the calculated BB contributions multiplied by a factor of 3. (a) CAM5-ATRAS model simulations. (b) MRI-ESM2 model simulations.






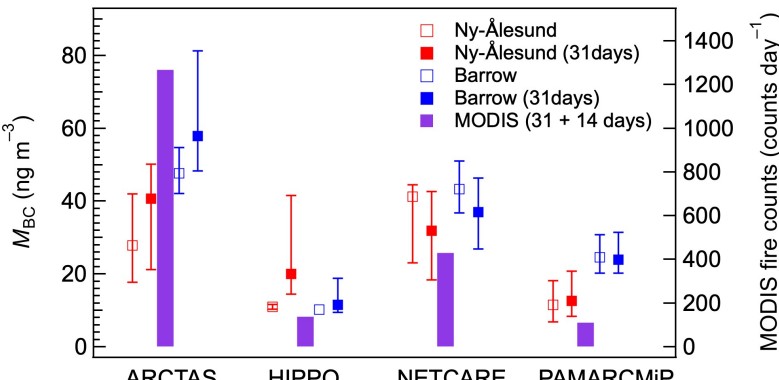

**Figure 6.** BC mass concentrations ($M_{BC}$) obtained by ground-based measurements at Ny-Ålesund, Svalbard (red), and Barrow, Alaska (blue). Median values of daily averages and 25–75% ranges are shown for the time periods of the individual aircraft experiments (open squares) and the 31-day period for which the median date was chosen to be the median date of the experiment (closed squares). Corresponding averaged fire counts (31 + 14 days) at latitudes north of 50°N are also shown (purple bars). Because the daily averaged $M_{BC}$ was obtained only on one day at Barrow for the HIPPO experiment period (two days), 25–75% range is not shown.





**(a)**

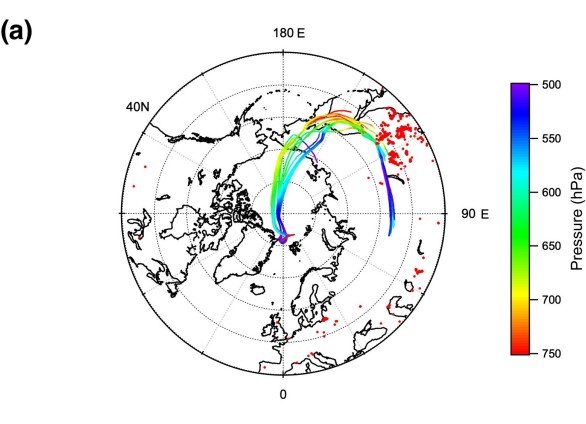

**(b)**

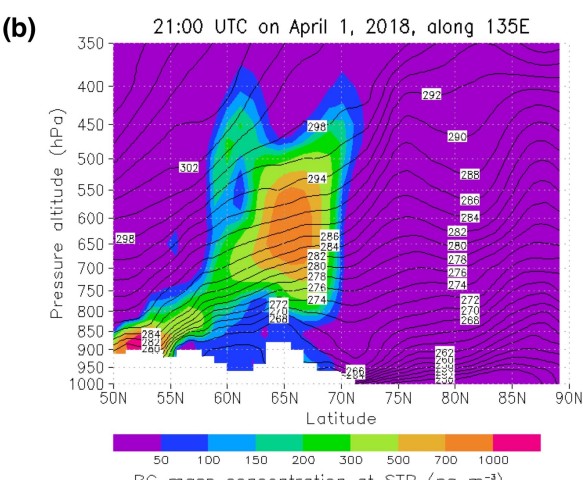

**(c)**

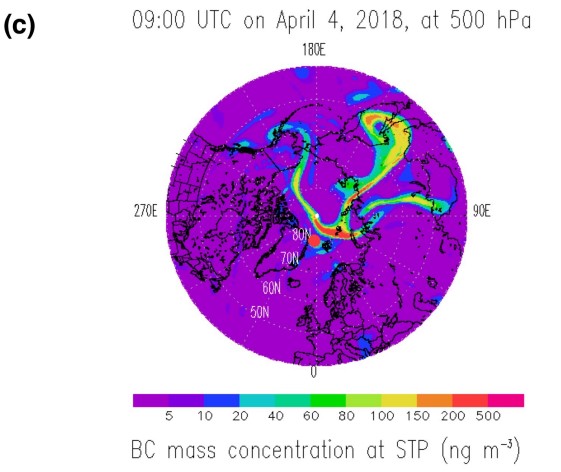

**Figure 7.** (a) Eight-day backward trajectories of air with enhanced BC mass concentrations ($M_{BC}$)
observed on 2, 3, and 4 April (the data points shown in red in Fig. 2a). Red points are MODIS-derived



hot spots observed during 25–27 March, 8 days before the aircraft observations of the enhanced $M_{BC}$. (b) Model calculated latitude–pressure altitude cross-section of $M_{BC}$ along the 135°E line (color) and potential temperature (contours) at 21 UTC on 1 April (MRI-ESM2 model). $M_{BC}$ values from BB origin are shown. (c) Horizontal distribution of $M_{BC}$ (BB origin) at 500 hPa level at 09 UTC on 4 April when

the enhanced $M_{BC}$ data were observed (MRI-ESM2 model). The red circle denotes the aircraft location when the enhanced $M_{BC}$ data were observed.






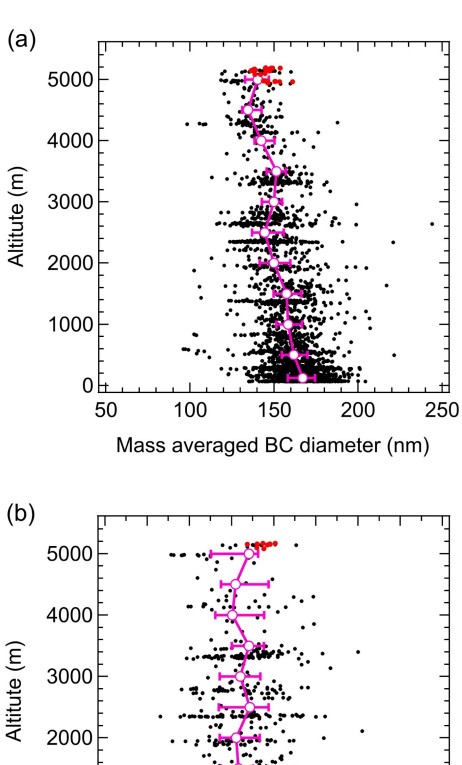

**Figure 8.** (a) Vertical profile of mass averaged diameters of BC ($D_{BC}$). Closed circles denote one-minute data that were calculated from one-minute BC mass concentrations ($M_{BC}$) and BC number concentrations using eq. (1). Median values and 25–75% ranges are also shown. Data points shown in red (enhanced $M_{BC}$ data points shown in red in Fig. 2a) were likely influences by biomass burning emissions. (b) Same as (a) but for the median shell-to-core diameter ratios for particles with a mass equivalent BC diameter (core diameter) of 180–192 nm.






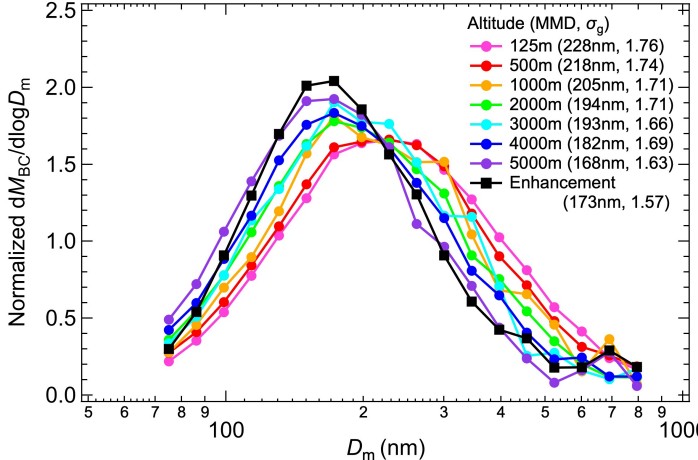

**Figure 9.** Averaged mass size distribution of BC in each altitude range. Values normalized by the total mass concentrations of BC (($dM_{BC}/dlogD_m$)/$M_{BC}$) are shown. The average size distribution for the enhanced $M_{BC}$ data is shown with black squares. The numbers shown in parentheses are the mass median diameter (MMD) and the geometric standard deviation ($\sigma_g$) for the lognormal fitted size distributions.


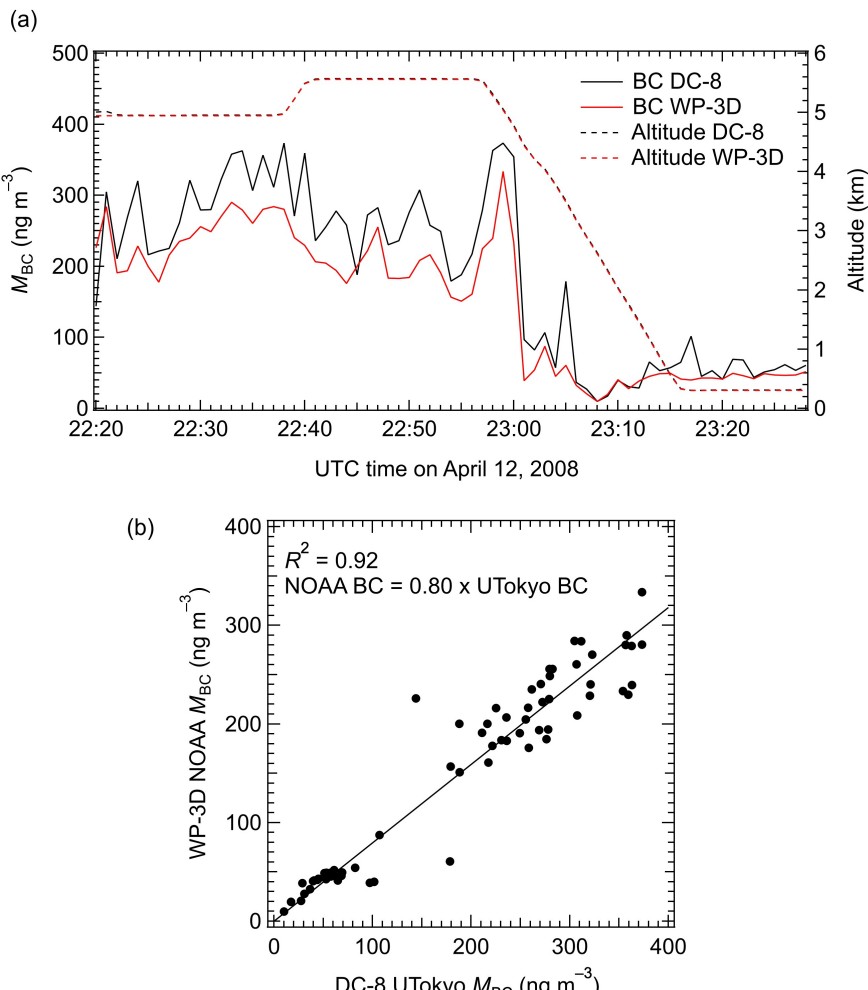

**Figure A1.** Comparison between $M_{BC}$ measurements made with the University of Tokyo SP2 during ARCTAS onboard the NASA DC-8 (black lines, Sahu et al., 2012) and NOAA SP2 measurements onboard the NOAA WP-3D (red lines, NOAA-ARCPAC experiment, Spackman et al., 2010). (a) Time series of one-minute data of $M_{BC}$ values (solid lines) and aircraft altitudes (dashed lines). (b) Scatter plot between these $M_{BC}$ values. The solid line is the least-squares fitted line forced through (0, 0).






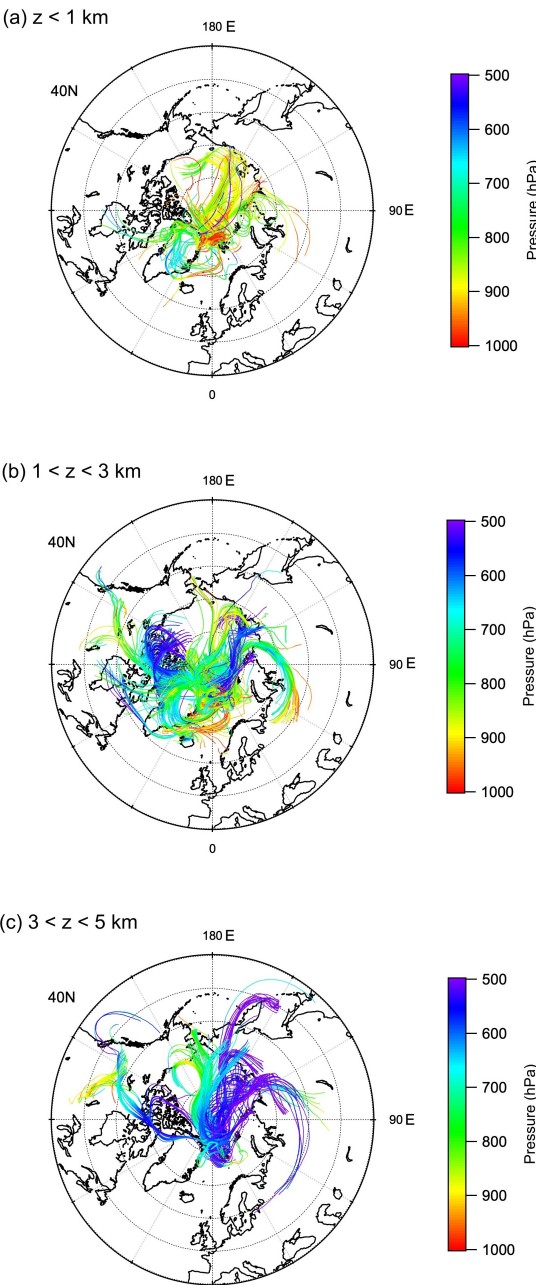

**Figure B1.** Six-day kinematic back trajectories of air parcels measured onboard the aircraft. Color scales denote the pressure altitude of air. (a) Trajectories starting from altitudes z < 1 km. (b) Same as (a) but for 1 < z < 3 km. (c) Same as (a) but for 3 < z < 5 km.

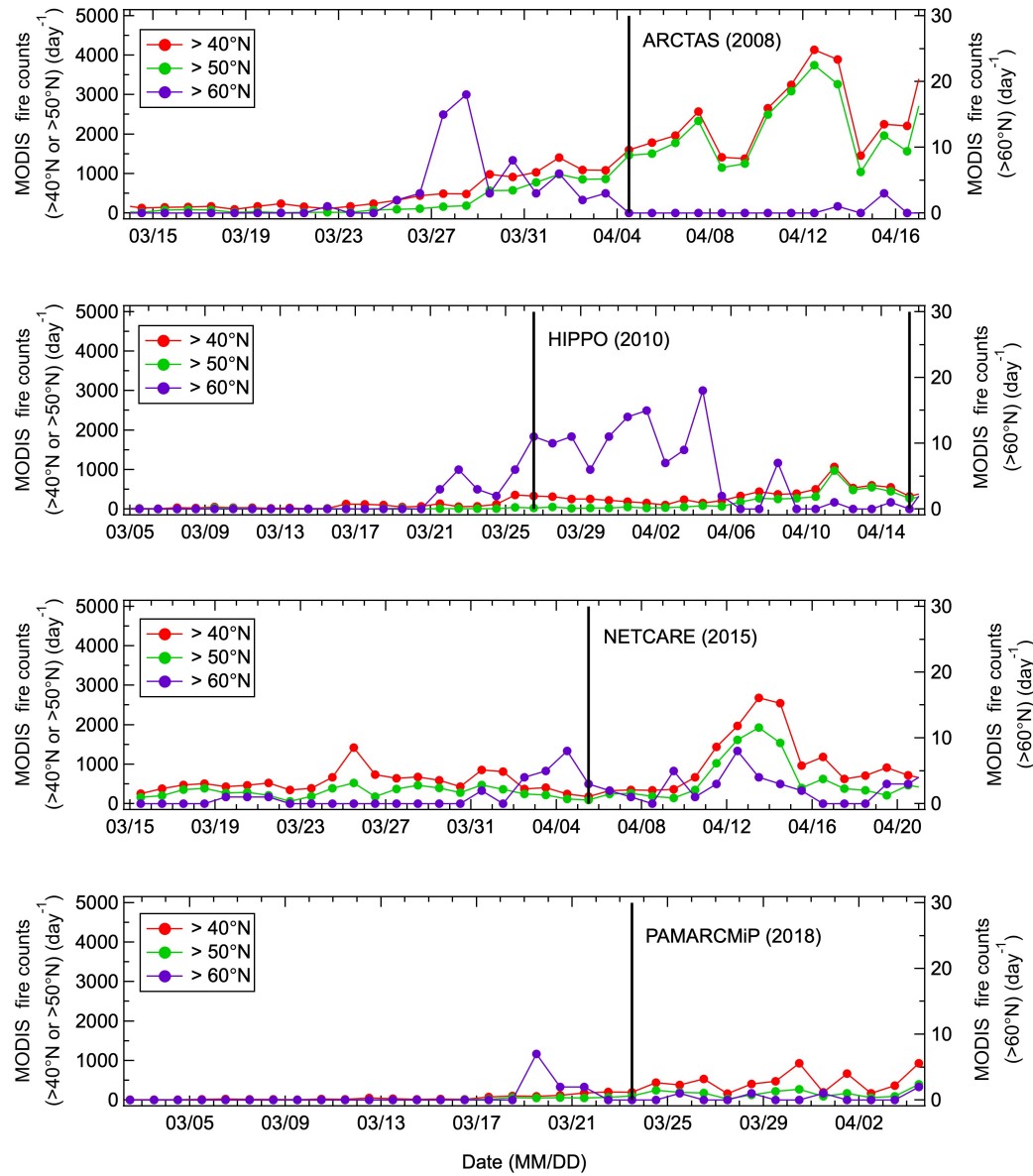

**Figure C1.** Time series of the daily averaged number of fire counts (counts day$^{-1}$) detected by the MODIS satellite. Vertical bars indicate the start dates of the individual aircraft-based experiments.



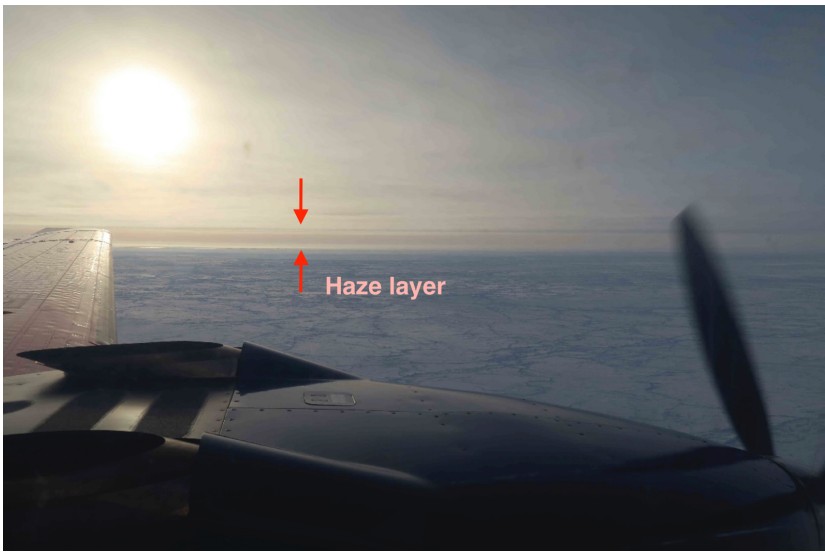


**Figure D1.** Photo taken from the research aircraft POLAR 5 on 3 April. A haze layer with reddish color, which may correspond to the plumes, was captured.






**Tables**

**Table 1.** Summary of research flights made during the PAMARCMIP experiment. The time is UTC.

| Flight | Date | Takeoff time | Landing time | Maximum altitude (m) |
|--------|------|--------------|--------------|----------------------|
| F01 | 3/23 | 11:53 | 12:47 | 1988 |
| F02 | 3/25 | 14:12 | 17:55 | 5131 |
| F03 | 3/26 | 11:53 | 15:41 | 5155 |
| F04 | 3/26 | 16:49 | 18:06 | 805 |
| F05 | 3/27 | 13:00 | 17:04 | 4987 |
| F06 | 3/28 | 11:00 | 13:05 | 2664 |
| F07 | 3/28 | 14:58 | 17:27 | 2401 |
| F08 | 3/30 | 07:42 | 13:07 | 4932 |
| F09 | 3/31 | 11:30 | 17:39 | 1362 |
| F10 | 4/2 | 10:07 | 14:14 | 5171 |
| F11 | 4/2 | 15:07 | 16:29 | 2213 |
| F12 | 4/3 | 07:53 | 11:54 | 5187 |
| F13 | 4/3 | 12:54 | 16:44 | 3324 |
| F14 | 4/4 | 07:51 | 11:50 | 4975 |



**Table 2.** Vertical profile of BC observed during the PAMARCMiP experiment. One-minute data were used for these statistics. The plume data were defined as those with $M_{BC} > 50$ ng m$^{-3}$ at STP.

| Altitude (m) | BC mass concentration ($M_{BC}$) (ng m$^{-3}$ at STP) | | Mass averaged BC diameter ($D_{BC}$) (nm) | | Shell-to-core diameter ratio | |
|---|---|---|---|---|---|---|
| | Median value | 25−75% range | Median value | 25−75% range | Median value | 25−75% range |
| 0−250 | 12.3 | 8.7−27.8 | 167.1 | 158.3−174.4 | 1.43 | 1.34−1.52 |
| 250−750 | 14.7 | 9.2−26.4 | 161.9 | 154.6−170.0 | 1.42 | 1.34−1.49 |
| 750−1250 | 8.4 | 4.1−16.0 | 158.4 | 151.4−166.8 | 1.40 | 1.33−1.46 |
| 1250−1750 | 8.0 | 5.4−12.1 | 157.4 | 149.7−166.2 | 1.43 | 1.35−1.50 |
| 1750−2250 | 6.7 | 2.7−13.9 | 150.0 | 141.5−159.7 | 1.42 | 1.34−1.53 |
| 2250−2750 | 8.8 | 3.4−14.7 | 144.5 | 136.9−155.7 | 1.49 | 1.34−1.58 |
| 2750−3250 | 13.0 | 8.1−27.5 | 150.0 | 142.9−154.9 | 1.44 | 1.34−1.54 |
| 3250−3750 | 17.7 | 9.7−28.0 | 151.5 | 145.6−157.2 | 1.48 | 1.40−1.55 |
| 3750−4250 | 13.4 | 9.5−17.3 | 142.4 | 138.4−150.4 | 1.40 | 1.32−1.55 |
| 4250−4750 | 9.5 | 5.6−15.3 | 134.5 | 131.2−142.6 | 1.42 | 1.35−1.58 |
| 4750−5250 | 22.8 | 11.4−80.9 | 140.3 | 132.7−146.9 | 1.48 | 1.30−1.53 |
| 4750−5250 (in the plumes) | 103.1 | 78.9−163.6 | 145.1 | 141.9−149.2 | 1.55 | 1.52−1.58 |











**Table 3.** Aircraft experiments made during the Arctic spring.

| Experiment | Year | Date | Location | Number of flights | SP2 operation | Reference |
|---|---|---|---|---|---|---|
| ARCTAS [a] | 2008 | 4–17 April | 66.50–89.97°N, 168.66–37.22°W | 7 | Univ. of Tokyo | Matsui et al., 2011 |
| HIPPO [a] | 2010 | 26 March, 15 April | 66.53–85.07°N, 149.88–147.24°W | 1, 1 | NOAA | Schwarz et al., 2013 |
| NETCARE | 2015 | 5–21 April | 66.73–83.53°N, 135.00°W – 14.63°E | 10 | AWI | Schulz et al., 2019 |
| PAMARCMiP | 2018 | 23 March– 4 April | 78.29–84.59°N, 23.95°W–19.57°E | 14 | Univ. of Tokyo | This study |

[a] Although measurements were made at latitudes south of 60°N, only data obtained at latitudes north of 66.5°N were used in this study. Dates and numbers of flights correspond to these Arctic measurements.









**Table 4.** Column BC amounts and biomass burning activities.

| Experiment | $COL_{BC}$ ($\mu g\ m^{-2}$) (>66.5°N) [a] | | $COL_{BC}$ ($\mu g\ m^{-2}$) (>80°N) [a] | | Averaged fire counts (counts day$^{-1}$) | Averaged GFAS BC emissions ($\mu g\ km^{-2}$ s$^{-1}$) | Averaged GFED BC emissions ($\mu g\ km^{-2}$ s$^{-1}$) |
|---|---|---|---|---|---|---|---|
| | Median value | 25–75% range | Median value | 25–75% range | | | |
| ARCTAS | 200 (4.2) | 116–351 | 190 (3.6) | 114–325 | 1281 (13.9) | 402.4 (5.86) | 377.3 (11.32) |
| HIPPO | 66 (1.4) | 48–94 | 75 (1.4) | 58–95 | 130 (1.4) | 17.5 (0.26) | 17.2 (0.51) |
| NETCARE | 132 (2.7) | 66–261 | 123 (2.3) | 86–179 | 469 (5.1) | 250.7 (3.65) | 137.7 (4.13) |
| PAMARC MiP | 48 (1.0) | 28–91 | 53 (1.0) | 34–101 | 92 (1.0) | 68.7 (1.0) | 33.3 (1.0) |

[a] $COL_{BC}$ is the column BC amount between 0 and 5 km.

Values in parentheses are relative to PAMARCMiP.











**Table 5a.** Model-calculated column BC amounts with the CAM5-ATRAS model.

| Experiment | $COL_{BC}(A)$[a], µg m$^{-2}$) | | | $COL_{BC}(B)$ (µg m$^{-2}$) | | | $COL_{BC}(C)$ (µg m$^{-2}$) | | |
|---|---|---|---|---|---|---|---|---|---|
| | AN | BB | AN+BB | AN | BB | AN+BB | AN | BB | AN+BB |
| ARCTAS | 53.9 | 7.9 | 61.8 | 42.9 | 10.0 | 52.8 | 66.1 | 21.1 | 87.2 |
| HIPPO | 56.3 | 1.1 | 57.5 | 62.7 | 1.5 | 64.3 | 70.6 | 1.4 | 72.0 |
| NETCARE | 40.4 | 4.7 | 45.1 | 39.5 | 3.5 | 43.0 | 64.6 | 7.9 | 72.6 |
| PAMARCMiP | 63.6 | 1.0 | 64.6 | 71.8 | 1.7 | 73.5 | 89.4 | 3.2 | 92.6 |

[a] $COL_{BC}$ is the column BC amount between 0 and 5 km.

[b] AN and BB stand for anthropogenic and biomass burning origins, respectively.

(A): median values along the flight tracks. (B): area-weighted averaged values within the latitudes and longitudes and the time period of the aircraft experiment shown in Table 3. (C): area-weighted averaged values within the entire region at latitudes north of 66.5°N for the time period of the aircraft experiment.












**Table 5b.** Model-calculated column BC amounts with the MRI-ESM2 model.

| Experiment | $COL_{BC}$(A)[a], µg m$^{-2}$) | | | $COL_{BC}$(B) (µg m$^{-2}$) | | | $COL_{BC}$(C) (µg m$^{-2}$) | | |
|---|---|---|---|---|---|---|---|---|---|
| | AN | BB | AN+BB | AN | BB | AN+BB | AN | BB | AN+BB |
| ARCTAS | 34.3 | 14.6 | 49.0 | 30.8 | 26.7 | 57.5 | 41.4 | 64.5 | 105.9 |
| HIPPO | 33.7 | 2.3 | 36.0 | 42.8 | 2.9 | 45.7 | 52.0 | 2.3 | 54.3 |
| NETCARE | 26.3 | 27.7 | 53.9 | 28.5 | 19.1 | 47.6 | 41.0 | 30.8 | 71.8 |
| PAMARCMiP | 47.9 | 10.7 | 58.7 | 49.1 | 9.9 | 59.0 | 53.2 | 18.1 | 71.3 |

[a] $COL_{BC}$ is the column BC amount between 0 and 5 km.

[b] AN and BB stand for anthropogenic and biomass burning origins, respectively.

(A): median values along the flight tracks. (B): area-weighted averaged values within the latitudes and longitudes and the time period of the aircraft experiment shown in Table 3. (C): area-weighted averaged values within the entire region at latitudes north of 66.5°N for the time period of the aircraft experiment.





**Table 6.** BC mass concentrations observed at the ground surface.

| Experiment | $M_{BC}$ at Ny-Ålesund (ng m$^{-3}$) | | $M_{BC}$ at Barrow (ng m$^{-3}$) | | Averaged fire counts (counts day$^{-1}$) |
|---|---|---|---|---|---|
| | Median value | 25–75% range | Median value | 25–75% range | |
| ARCTAS | 41 (3.2) | 21–50 | 58 (2.4) | 48–81 | 1267 (11.4) |
| HIPPO | 20 (1.6) | 15–42 | 12 (0.5) | 9–19 | 139 (1.3) |
| NETCARE | 32 (2.5) | 18–43 | 37 (1.5) | 27–46 | 430 (3.9) |
| PAMARCMiP | 13 (1.0) | 8–21 | 24 (1.0) | 20–31 | 111 (1.0) |

1195

Medians and 25–75% ranges were calculated for 31-day periods with a median date chosen to be the median date of the experiment given in Table 3. Fire counts were averaged for latitudes north of 50°N.