# Peer review of "Arctic black carbon during PAMARCMiP 2018 and previous aircraft experiments in spring"

_Atmospheric Chemistry and Physics, 2021_

## Author Comment (AC1)

**Responses to the Reviewers' Comments**

Title: Arctic black carbon during PAMARCMiP 2018 and previous aircraft experiments in spring MS No.: acp-2021-349 MS type: Research article

The co-authors and I very much appreciate the constructive comments on this manuscript by the reviewers. The comments have been very thorough and useful in improving the manuscript. We have taken them fully into account in revising it. This file contains one-by-one responses to these comments.

**Corresponding author**

Sho Ohata Institute for Space-Earth Environmental Research, Nagoya University Furo-cho, Chikusa-ku, Nagoya, Aichi, 464-8601, Japan. sho.ohata@isee.nagoya-u.ac.jp

**Responses to Anonymous Referee #1**

>> This manuscript provides a detailed analysis of observations of black carbon (BC) obtained during the PAMARCMIP research flights in the Arctic in 2018, as well as interpretations of data from previous experiments. The manuscript is well written and clear, and provides insight regarding the sources, characteristics, and variability of BC in the Arctic. The authors convincingly argue that inter-annual variability in BC abundance is largely governed by biomass burning (BB) amount and transport, while the anthropogenic component of BC is much less variable. This subject is of interest to ACP readers, and the work is of high quality. I recommend publication after minor revisions.

Answer 1-1: Thank you for the positive comments.

>> There are two primary areas I would like to see addressed in a revised manuscript. First, the authors compare BC column loadings between four different field programs and two climate models with nudged meteorology. The authors correctly note that model measurement comparison is affected by location mismatches between the measurements and model-simulated transport from biomass burning events. They address this concern by comparing larger spatial averages from the model domain with the measurements (Fig.5). However, they ignore vertical mismatches by limiting the comparison between model and measurement to the 0-5 km range reachable by the PAMARCMIP aircraft. Vertical distribution of emitted smoke is extremely difficult to simulate in models, due to considerable uncertainty in the initial injection height of the smoke as well as uncertainty in the vertical lifting during subsequent long-range transport. I would like to see the authors extend their analysis of the HIPPO, ARCTAS, and NETCARE data to the highest altitudes reached on those campaigns, as well as examine the sensitivity of the model simulated BC column amounts to different choices in integration height (e.g., 0-5 km, 0-7 km, 0-10 km).

Answer 1-2: In the revised manuscript, model-calculated vertical profiles of BC mass concentrations ( $M_{BC}$ ) have been compared with observations for entire altitude ranges in which observed data are available, namely, up to 10.5, 11.0, 5.5, and 5.0 km for ARCTAS, HIPPO, NETCARE, and PAMARCMiP, respectively (Sect. 4.3 and Appendix D). Also, comparisons between model calculations and observations of BC

column amounts ( $COL_{BC}$ ) for different altitude ranges (0–5 km, 0–7 km, and 0–10 km) have been described in these sections, as the reviewer suggested. We have found that when model-calculated  $COL_{BC}$  (0–5 km) generally well agree with the observations (HIPPO and PAMARCMiP; Fig. 5), reasonable agreements were found for  $M_{BC}$  values throughout the altitudes (Fig. D1a), while when model calculations underestimated  $COL_{BC}$  (0–5 km) they underestimated  $M_{BC}$  values at all altitudes. These results indicate that the model underestimations in  $COL_{BC}$  (0–5 km) values for high BB activity years (ARCTAS and NETCARE) were not due to vertical mismatches of enhanced  $M_{BC}$  layers. This conclusion is also consistent with analyses on  $COL_{BC}$  values for 0–10 km, namely, the features found for  $COL_{BC}$  values are similar between 0–5 and 0–10 km (Fig. D1b).

>> Second, the discussion of BC removal by precipitation could be sharpened. How well models simulate this removal is absolutely critical to their representation of BC abundance following long-range transport. The Arctic is a challenging environment to simulate removal due to the dominant role of mixed-phase clouds in this process. In the manuscript, the authors attempt to say something about transport efficiency by comparing observed dBC/dCO ratios following transport with those prescribed for the BB emissions in the models. This seems like an apples-to-oranges comparison. More important would be, how does the model dBC/dCO ratio compare with that from the trajectories? And how does the final dBC/dCO ratio compare with that from the measurements? Further analysis along these lines may help explain whether the model and measurement ratios differ because of errors in the emission ratios at the fire locations, or because of scavenging of BC during transport. Some more exploration of these issues would be welcome, as it's key to improving model representations of BC abundance in the Arctic.

**Answer 1-3:** In Sect. 5.1 of the revised manuscript, we have added estimation of transport efficiency of BC ( $TE_{BC}$ ) for the biomass burning (BB) plumes using the MRI-ESM2 model. Because this model calculation did not take CO into account, it was not possible to estimate dBC/dCO ratio using the model as suggested by the reviewer. Instead, we have estimated  $TE_{BC}$  along air parcel trajectories using the model-calculated removal flux of BC. We have found that the estimated  $TE_{BC}$  value was about 0.4–0.5, which is consistent with the  $TE_{BC}$  value estimated from dBC/dCO analysis based on observation and GFED emission ratio (0.58). From the model calculation, we have also found that most of the wet removals of BC took place upon lifting in association with

the passage of the cold frontal system and  $TE_{BC}$  changed little after air parcels reached in the free troposphere. In the revised manuscript, we have added these descriptions. On the other hand, we have not removed the discussion on dBC/dCO based on observation and GFED emission ratio, because we believe it is still useful to estimate  $TE_{BC}$  for BB plumes observed during PAMARCMiP campaign by this method for comparison with previously reported  $TE_{BC}$  values.

>> Technical corrections:

Line 145: Please spell out "DMT" and provide the country.

**Answer 1-4:** Because "Droplet Measurement Technology (DMT)" has appeared earlier for the explanation of the single-particle soot photometer (SP2), in the revised manuscript we have remained it as it is.

>>Line 319: Replace "larger increases" with "larger values".

Answer 1-5: We have corrected it, as the reviewer suggested.

>>Table 6: What are the values in parentheses?

**Answer 1-6:** Values in parentheses are relative to PAMARCMiP value. We have added this description in Table 6.

>> References: Please ensure that all journal names are abbreviated following Copernicus guidelines.

**Answer 1-7:** We have corrected several abbreviations of the journal names, following the Copernicus guidelines.

>> Figures: Please review all figures for compatibility for color-impaired readers. For example, vary line types and thicknesses and symbol types, and choose a color pallet that is more easily discernable. I know several scientists with this impairment and reading figure is often a problem for them. Thank you.

**Answer 1-8:** We have modified figures 1, 2, 3, 5, 6, 9, and C1 so that they can be more easily discernable for all readers. Thank you for the suggestion.

 $\mathbf{5}$

**Responses to Anonymous Referee #2**

>> The manuscript documents an Arctic aircraft campaign for measuring the amount and some properties of the black carbon aerosol. As three dimensional measurements are sparse, but important for understanding the Arctic atmospheric composition this is a very relevant contribution. The authors show additionally to the measurement data, data from two chemistry transport models, a trajectory analysis and satellite detected fire hot spots to investigate in the importance of the biomass burning aerosol for this, but also for past campaigns. The study is well presented, however I think it has some shortcomings by not exploring the modelling results sufficiently and relying only on fire hot spots counts on a northern mid and high latitude region.

**Answer 2-1:** Thank you for the constructive comments. In the revised manuscript, we have further investigated model results including vertical distributions and biomass burning/anthropogenic BC emissions, as described in the following answers.

>> Another aspect which should investigated, is a comparison of all the measurements (including station data), which are based on SP2 method with black carbon derived from other methods on nearby locations. For example eBC timeseries taken at Zeppelin station could be added.

Answer 2-2: The main focus of this study is on year-to-year variations of vertical profiles of BC (column amounts of BC), especially from the viewpoint of year-to-year variations of biomass burning (BB) activities. BC mass concentration ( $M_{BC}$ ) data at the ground stations (Ny-Ålesund and Barrow) during the 4 aircraft-based campaign periods are included in Sect. 4.4 to examine the difference of year-to-year variations between the free troposphere and ground surface. Therefore, more detailed analysis of time series of  $M_{BC}$  at Arctic ground stations is beyond the scope of this study. Seasonal variations, long-term trends, and sources of BC observed at the ground stations in the Arctic will be discussed in a separate paper.

>> The authors assume the models might underestimate the BB emission which are observed at the beginning of April. Instead of an extensive comparison to the fire hot spots I think following is missing:

There is no detailed explanation of how large the biomass burning and the anthropogenic emissions are in both models (maybe split for mid and high latitudes). Also the BC lifetime for both modelling approaches should be given.

Answer 2-3: We have added Fig. 3b to show the biomass burning (BB) and anthropogenic (AN) BC emissions used in the CAM5/ATRAS model for latitudes >50°N and >60°N. The AN BC emissions slightly decrease with years and no clear correspondence is found in year-to-year variations between AN BC emissions and observed BC column amount. BC emissions used for the MRI-ESM2 model are generally similar to those used for the CAM5/ATRAS model, although AN emissions for >60°N are generally smaller. We have added these descriptions in Sect. 4.2. The BC lifetime estimated using MRI-ESM2 at latitudes north of 66.5°N in April is 11.9  $\pm$  2.8 days. An annual average for Arctic BC is 12.1 and 11.5 days for MRI-ESM2 and CAM5-ATRAS simulations, respectively. We have also added these descriptions in Sect. 4.2 in the revised manuscript.

>> L263: The measurements of the PAMARCMIP campaign are given as average concentrations or a summary showing the vertical distribution. A time series of the aircraft data and the model results (showing both anthropogenic and biomass burning) would give a better insight in the variability of the measurements.

Answer 2-4: In general, time series of model calculations (with large time steps and large grid scales) are not directly comparable with time series of aircraft data, and thus median  $M_{\rm BC}$  and  $COL_{\rm BC}$  values of observation data are compared with model calculations in this study. For distinctive observation data such as relatively high  $M_{\rm BC}$  values around 5 km altitude (BB plumes), a more detailed analysis of model-calculated  $M_{\rm BC}$  has been performed for interpretation of the observations (i.e., transport efficiency of BC; Sect. 5.1). In the revised manuscript, we have also added Fig S1 in the supplement to show a time series of aircraft data that includes enhanced  $M_{\rm BC}$  values influenced by BB plumes.

Answer 2-5: In the revised manuscript (Sect. 4.3 and Appendix D), model-calculated

>> In Figure 2, which shows the vertical distribution also the modelled concentrations should be added.

vertical profiles of  $M_{BC}$  have been compared with observations for entire altitude ranges in which observed data are available, namely, up to 10.5, 11.0, 5.5, and 5.0 km for ARCTAS, HIPPO, NETCARE, and PAMARCMiP, respectively (Fig. D1).

>> Figure 7, which shows the biomass burning plume could be made more readable by showing the location of the transect on the map and removing the violet regions for the low or zero concentrations. Additionally the anthropogenic contribution could be shown.

Answer 2-6: The aircraft location when the enhanced  $M_{BC}$  data were observed is denoted by a red circle in Fig. 7c. We prefer to use the violet color for lowest  $M_{BC}$ values for technical reasons in this figure. Anthropogenic contribution was small and this has been mentioned in Sect. 5.1 in the revised manuscript.

>> Figure 4 shows the fire counts, I wonder why not BC emissions (as total mass emitted) have been used and the flight tracks could be added to get a better impression which fire sources potentially could influence the campaign.

Answer 2-7: As discussed in Sect. 4., BB BC emissions used for the models could be largely underestimated, so here we show the fire counts, which are more basic indicators of BB activities. As described in Answer 2-3, BB and AN BC emissions used for the CAM5/ATRAS model (latitudes >50°N and >60°N) are shown in Fig. 3b in the revised manuscript. The flight tracks have not been added to Fig. 4 for visibility reasons. The year-to-year variation of fire counts in northern high latitudes (45–60°N) in western and eastern Eurasia (around 30–50°E and 100–130°E, respectively) potentially influenced the year-to-year variation of BC column amount in the Arctic (Sect. 4.2)